# Involvement of the Acyl-CoA binding domain containing 7 in the control of food intake and energy expenditure in mice

Damien Lanfray[1]*, Alexandre Caron[1], Marie-Claude Roy[1], Mathieu Laplante[1], Fabrice Morin[2,3], Jérôme Leprince[2,3], Marie-Christine Tonon[2,3], Denis Richard[1]*

[1]Centre de Recherche de l'Institut Universitaire de Cardiologie et de Pneumologie de Québec, Université Laval, Québec, Canada; [2]Laboratory of Neuronal and Neuroendocrine Differentiation and Communication, Institut National de la Santé et de la Recherche Médicale, Mont-Saint-Aignan, France; [3]Institute for Research and Innovation in Biomedicine, Normandy University, Mont-Saint-Aignan, France

**Abstract** Acyl-CoA binding domain-containing 7 (*Acbd7*) is a paralog gene of the diazepam-binding inhibitor/Acyl-CoA binding protein in which single nucleotide polymorphism has recently been associated with obesity in humans. In this report, we provide converging evidence indicating that a splice variant isoform of the *Acbd7* mRNA is expressed and translated by some POMC and GABAergic-neurons in the hypothalamic arcuate nucleus (ARC). We have demonstrated that the ARC ACBD7 isoform was produced and processed into a bioactive peptide referred to as nonadecaneuropeptide (NDN) in response to catabolic signals. We have characterized NDN as a potent anorexigenic signal acting through an uncharacterized endozepine G protein-coupled receptor and subsequently via the melanocortin system. Our results suggest that ACBD7-producing neurons participate in the hypothalamic leptin signalling pathway. Taken together, these data suggest that ACBD7-producing neurons are involved in the hypothalamic control exerted on food intake and energy expenditure by the leptin-melanocortin pathway.

*For correspondence: lanfray.
damien@gmail.com (DL); Denis.
richard@criucpq.ulaval.ca (DR)

**Competing interests:** The authors declare that no competing interests exist.

## Introduction

The understanding of the complex brain controls of food intake and energy expenditure is essential to unravel the causes of excess fat deposition and to envision effective strategies to prevent or reverse obesity. The control of food intake and energy expenditure, hence the regulation of energy balance, is assured by interconnected neurons of various brain regions, which include the hypothalamus (*Morton et al., 2006*, *Schwartz et al., 2000*, *Richard, 2015*). Among all the hypothalamic nuclei, the arcuate nucleus (ARC), which hosts proopiomelanocortin (POMC) and agouti-related peptide (AgRP) / neuropeptide (NPY) - producing neurons, has emerged as a prominent structure in the control of both food intake and energy expenditure (*Krashes et al., 2013*, *Mayer and Belsham, 2009*, *Luquet et al., 2005*, *Gropp et al., 2005*). POMC and NPY/AgRP neurons are major constituents of the melanocortin system, which is recognized to genuinely govern energy balance regulation (*Adan et al., 2006*, *Butler, 2006*, *Cone, 2006*, *De Jonghe et al., 2011*, *Ellacott and Cone, 2006*, *Xu et al., 2011*). POMC neurons release α-melanocyte-stimulating hormone (α-MSH), which induces hypophagic and thermogenic effects, mainly through the stimulation of the melanocortin-4 receptor (MC4R) (*Adan et al., 2006*, *Butler, 2006*, *Cone, 2006*). AgRP has been described as an inverse (*Chai et al., 2003*, *Haskell-Luevano and Monck, 2001*) or biased (*Buch et al., 2009*) agonist of this receptor. In this context, the understanding of the mechanisms whereby the melanocortin regulate energy homeostasis is critical to unravel the "physiopathology" of obesity.

**eLife digest** Obesity is an increasingly common problem worldwide. To treat it effectively, we must understand how the body controls how much food a person consumes and how much energy they expend. The hypothalamus is one region of the brain that plays a critical role in regulating this energy balance. Some of the neurons in the hypothalamus can change their activity when they detect satiety hormones including the leptin, which is produced by fat cells and suppresses appetite. However, it is not clear exactly how the neurons respond to leptin and other energy-related signals.

Recent studies have linked the gene that encodes a protein called ACBD7 with obesity, and showed that it is one of the genes that is overexpressed in neurons that are sensitive to leptin. Now, Lanfray et al. have discovered a population of neurons that produce a new variant of the protein in the hypothalamus of mice. When this protein variant matures, it can be cut down to form a small protein-like molecule called NDN. Further experiments showed that leptin stimulates the production of both the new ABCD7 variant and NDN.

Lanfray et al. then injected mice that had been denied food for a several hours with NDN. The injected mice ate less than untreated mice, and burn more energy.

NDN appears to form part of the signaling pathway through which leptin signals to the hypothalamus to control appetite. In the future, creating mice in which the activity of the gene that encodes ACBD7 can be easily disrupted could help to reveal more about how the hypothalamus helps to control energy balance.

Evidence has accumulated in recent years, which suggests that endozepines (EZs) could play a role in the regulation of energy balance. EZs are described as glial endogenous peptides (*Tonon et al., 1990*), which are derived from the diazepam-binding inhibitor/Acyl-CoA binding protein (*Dbi/Acbp*) gene (*Mogensen et al., 1987*, *Ferrero et al., 1984*, *Corda et al., 1984*), and which include DBI/ACBP itself, the triakontatetraneuropeptide (TTN) and the octadecaneuropeptide (ODN). EZs were initially characterized for their ability to displace benzodiazepines from their binding sites (i.e. the $\gamma$-aminobutyric acid type A receptor (GABA$_A$-R) and the translocator protein (TSPO) (*Ferrero et al., 1984*, *Slobodyansky et al., 1989*). Recently, ODN emerged as endogenous modulator of the brain melanocortin signalling pathway and as a potential actor in the hypothalamic regulation of energy homeostasis (*Lanfray et al., 2013*), by acting via an uncharacterized G protein-coupled receptor (GPCR) (*do Rego et al., 2007*). Further supporting the role of *Dbi/Acbp* products in energy balance, a study conducted in humans by Comuzzie and collaborators (*Comuzzie et al., 2012*), indicates that *DBI/ACBP* single nucleotide polymorphism (SNP) was associated with obesity, supporting the relevance of the DBI/ACBP production in energy homeostasis. Interestingly, Comuzzie and collaborators (*Comuzzie et al., 2012*) also identified a SNP in a well-conserved paralog gene of the *DBI/ACBP*, referred as Acyl-CoA binding domain containing 7 (*ACBD7*), as associated to obesity in humans (*Comuzzie et al., 2012*), suggesting that *ACBD7* could be involved in energy homeostasis through encoding products related to EZs.

The present study aimed at investigating the role of *Acbd7* in energy homeostasis in mice. We hypothesised a role for *Acbd7* and its encoded products in the control of energy intake and energy expenditure through modulating the networks genuinely involved in energy homeostasis, including the leptin-melanocortin circuit.

## Results

### *Acbd7* mRNA sequence and hypothalamic expression of ACBD7

*In silico* analysis indicates that the *Acbd7* gene is a well-conserved paralog gene of the *Dbi/Acbp* (*Figure 1—figure supplement 1a,b*), that could also lead to the production of bioactive fragments in several vertebrate species, including human and rodent (*Figure 1—figure supplement 1a,b,c*). We first verified the sequence of *Acbd7* mRNA in the murine brain and demonstrated a prevalent sequence (*Figure 1a*) that differs from that already described (*Acbd7*-001) in the NCBI database (http://www.ncbi.nlm.nih.gov/nuccore/NM_030063.2). The observed splice variant contained 3

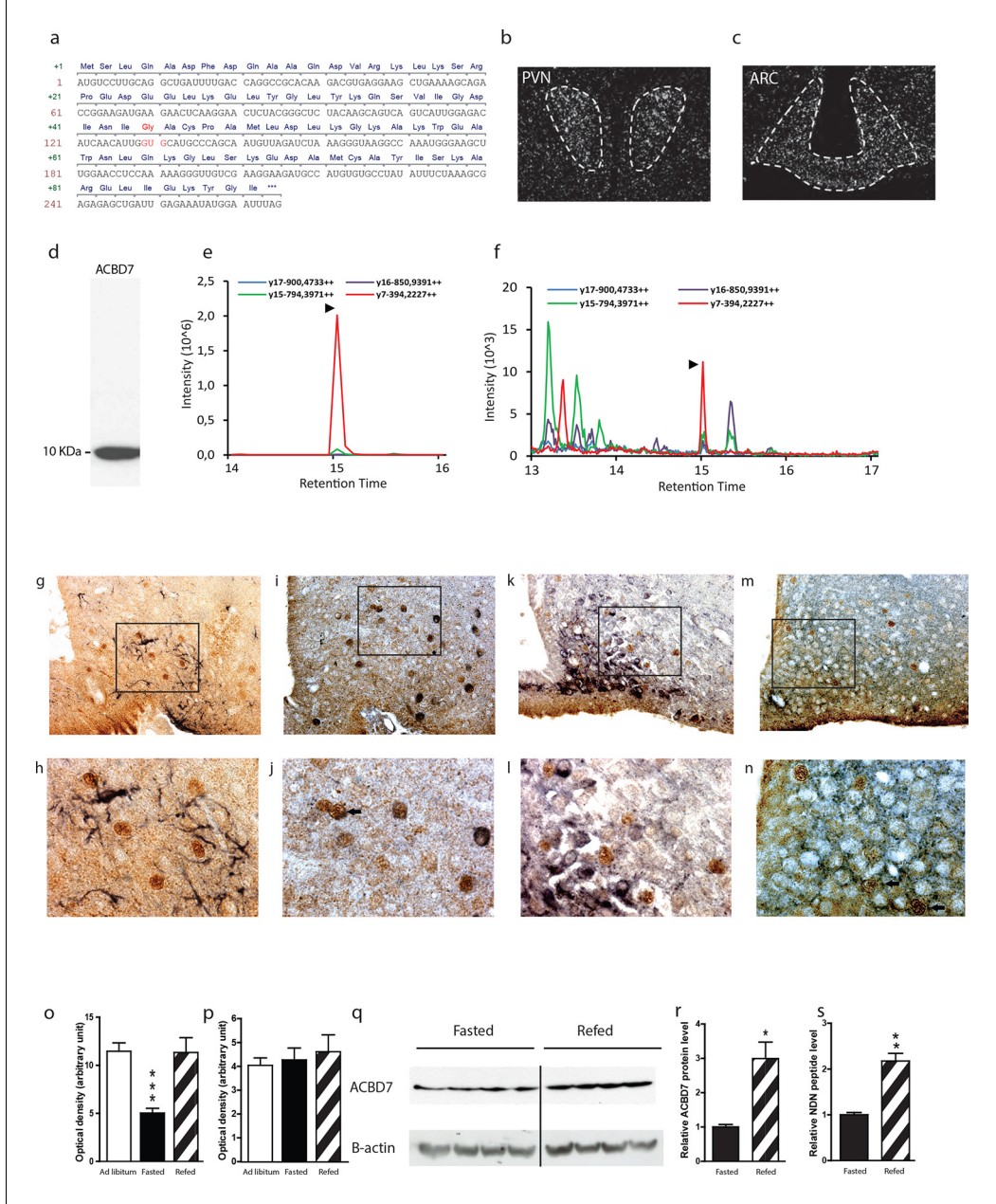

**Figure 1.** *Acbd7* mRNA sequence, distribution and processing in the mouse brain. (**a**) Sequence of the *Acbd7* mRNA open reading frame and the expected encoded protein. Additional codon and resulting amino acids are marked in red. (**b,c**) *In situ* hybridization showing *Acbd7* mRNA distribution in mouse hypothalamus. Abbreviations: PVN, paraventricular nucleus; ARC, arcuate nucleus. Dashed lines indicate the boundaries of nuclei (**d**) Western blot performed on mediobasal hypothalamic homogenate using an ACBD7-specific antibody. MRM-MS analysis profile obtained using (**e**) synthetic NDN or (**f**) hypothalamic protein lysate as template. (**g–j**) Mediobasal hypothalamic sections labeled with ACBD7-specific antibody (**g–m**; brown labelling) and GFAP antibody (**g,h**; black labelling), POMC antibody (**i,j**; black labelling), NPY antibody (**k,l**; black labelling) or VGAT antibody (**m,n**; black labelling). (**h,j,l,m**) Higher magnification views of scare panels defined in (**g**), (**i**), (**k**) and (**m**) respectively. Apparent co-labelling is indicated by an arrow (**j,n**). (**o,p**) *In situ* hybridization analysis of *Acbd7* mRNA levels in the ARC (**o**) and PVN (**p**) of *ad libitum*-fed mice, 18 hr-food deprived mice or 18 hr-food deprived mice having access to food 2 hr before sacrifice. Data were compared to *ad libitum*-fed mice as control (*n*=8). Data are expressed as mean ± SEM. One-way ANOVA, followed by a post-hoc multiple comparison Bonferroni test: ***p<0.001. (**q,r**) Western blot analysis of hypothalamic protein lysates from 18 hr-fasted mice or mice having access to food 6 hr before sacrifice, performed using ACBD7 and β-actin antibodies. (**r**) Quantification of the

*Figure 1 continued on next page*

*Figure 1 continued*

relative ACBD7 protein levels performed using β-actin signal as the loading control. (**s**) MRM-MS analysis of hypothalamic NDN peptide levels performed using exogenous peptide as control (*n*=4). Data are expressed as mean ± SEM. Unpaired Student's *t* test: \*\*p< 0.01; \*\*\*p<0.001.

The following source data and figure supplements are available for figure 1:

**Source data 1.** Synthetic NDN MRM-MS profile.
**Source data 2.** Hypothalamic lysate MRM-MS profile.
**Source data 3.** Impact of body energy status on ARC Acbd7 mRNA levels mRNA levels.
**Source data 4.** Impact of body energy status on PVN Acbd7 mRNA levels.
**Source data 5.** Impact of body energy status on hypothalamic ACBD7 levels.
**Source data 6.** Impact of body energy status on hypothalamic NDN levels.
**Figure supplement 1.** *Acbd7* mRNA sequence and conservation.
**Figure supplement 2.** Relative Acbd7 mRNA levels in mice brain structures.
**Figure supplement 3.** ACBD7 labelling observed without prior colchicine or 3-MA treatment.

additional nucleotides at the splice junction of exons 2 and 3, (*Figure 1a*), thereby predicting the insertion of a single glycine between the Ile[43] and Ala[44] of the originally expected 88-amino acid-encoded ACBD7-001 protein (herewith referred as ACBD7$_{88}$—*Figure 1—figure supplement 1d*) and the production of a 89-amino acid-containing protein isoform (herewith referred as ACBD7$_{89}$).

Using *in situ* hybridization (ISH), we demonstrated the presence of *Acbd7* mRNA, in several brain nuclei, including the paraventricular hypothalamic nucleus (PVH) (*Figure 1b*), ARC (*Figure 1c*), piri-form cortex, nucleus accumbens, median preoptic nucleus, supraoptic nucleus, medial amygdala, and hippocampus (*Figure 1—figure supplement 2*). Significant hybridization signal was observed in the ARC and PVH (*Figure 1b,c*). No labelling was observed when the *Acbd7* cDNA specific sense riboprobe was used (data not shown), confirming the specificity of the *Acbd7* riboprobe.

We also demonstrated that hypothalamic cells could translate the *Acbd7* mRNA into ACBD7$_{89}$ and to process it into a 19-amino-acid fragment (ACBD7$_{89 (34-52)}$) that we named nonadecaneuro-peptide (NDN). Western blot experiments, using immuno-purified ACBD7 specific antibody, revealed the presence in the mediobasal hypothalamus (MBH) of a single band of 10 kDa, (*Figure 1d*), confirming that ACBD7 was endogenously produced in the MBH. In order to establish the ability of hypothalamic cells to process ACBD7$_{89}$ into NDN, we performed multiple reaction monitoring mass spectrometry (MRM-MS) investigations. This protocol performed on mice MBH extracts, using synthetic NDN as control (*Figure 1e*), indicated that endogenous NDN was found in the hypothalamic lysates (*Figure 1f*), thus confirming that the ACBD7$_{89}$ isoform is produced but also processed in the mouse brain.

Immunohistochemical analysis performed using the ACBD7-specific antibody revealed that ACBD7 labelling was not observed without colchicine or 3-methyladenin (3-MA) pre-treatment (*Figure 1—figure supplement 3*). Double immunostaining performed on brain sections of colchicine-treated mice revealed that ACBD7 was not stained in glial cells (GFAP) in the MBH (*Figure 1g,h*). We also observed that ACBD7 was stained in some ARC POMC neurons (*Figure 1i,j*) and some vesicular GABA transporter (VGAT) positive neurons (*Figure 1m,n*) in 3-MA-treated mice, indicating that ACBD7 is predominantly produced by neurons in the MBH. However, investigation performed on colchicine-treated mice revealed no apparent ACBD7 staining in NPY neurons (*Figure 1k,l*).

Since ACBD7$_{89}$ appeared produced and processed by neurons of the ARC, whose involvement in energy homeostasis has been acknowledged, we investigated whether the whole body energy varia-tions could affect the *Acbd7* mRNA levels as well as the production of ACBD7 and its maturation

into NDN. We observed that 18 hr of fasting was sufficient to reduce *Acbd7* mRNA levels in the ARC of control mice by about 50%, whereas refeeding (2h) restored *Acbd7* expression (*Figure 1o*). Notably, no effect of fasting or even refeeding challenge was observed on Acbd7 mRNA levels in the paraventricular nucleus (PVN) (*Figure 1p*). Additionally, western blot analysis performed on MBH extract using fasted mice as controls, indicated that refeeding (6h) increased the protein levels of ACBD7 by about 3 times (*Figure 1q,r*), suggesting that both *Acbd7* mRNA and ACBD7 protein levels in the ARC correlated with the body energy status. Thereafter, we assessed by MRM-MS the impact of refeeding challenge on the maturation of $ACBD7_{89}$ into NDN, within the MBH. Consistently with our previous results, the MBH levels of NDN increased 2 fold after 2 hr of refeeding (*Figure 1s*), suggesting that production of $ACBD7_{89}$ and its processing into NDN was influenced by the energy status.

## Effects of NDN on food intake and energy expenditure

To address the impact of ACBD7 maturation products on feeding behaviour, we examined the effect of intracerebroventricular (icv) injections of graded doses of NDN on food intake and compared it with that of the $ACBD7_{88}$-derived peptide $ACBD7_{88\ (34-51)}$. We also tested the effect of the C-terminal octapeptide of NDN, i.e. the $NDN_{(12-19)}$ ($ACBD7_{89\ (45-52)}$). The injections were done in fasted mice. We observed that all injected compounds led to anorexigenic effects that were seen as early as 30 min after injection. However, dose/response experiments indicate that the optimal concentrations to cause hypophagia differed for each peptide (*Figure 2d*). Indeed, a potent inhibition of food intake was observed in mice administrated with 10 ng ($5.1\ 10^{-12}$ mol) of NDN (*Figure 2a*), while this effect was only achieved when 100 ng ($5.3\ 10^{-11}$ mol) of the $ACBD7_{88\ (34-51)}$ fragment were icv-injected (*Figure 2b*). Consistently, $NDN_{(12-19)}$, the common C-terminal octapeptide of $ACBD7_{88\ (34-51)}$ and NDN was efficient for doses as low as 5 ng ($5.6\ 10^{-12}$ mol) (*Figure 2c*), suggesting that the biological activity of the ACBD7-derived peptides is ensured by this well-conserved fragment. Furthermore, we observed that the anorexigenic effect induced by $NDN_{(12-19)}$ was not present 24 hr after the icv injection (*Figure 2d*). Interestingly, comparison of the efficiency of each peptide, 3 hr after injection, revealed that NDN as well as $NDN_{(12-19)}$ are both 10 times more potent than $ACBD7_{88\ (34-51)}$, in reducing food intake by 50% (*Figure 2d*). We also monitored the energy expenditure after acute NDN (10 ng) icv injection in fasted mice. NDN increased the energy expenditure in mice (*Figure 2e,f*), while increasing the $O_2$ consumption (*Figure 2g*) and $CO_2$ production (*Figure 2h*), without affecting the respiratory quotient (*Figure 2i*) or the physical activity pattern (*Figure 2j*), over 6 hr following the icv injection. Consistently with those results, experiments performed on fasted mice also revealed that acute icv injection of NDN (10 ng) led to significant increase in the expression of uncoupling protein 1 (*Ucp1*) mRNA level in the interscapular brown adipose tissue (iBAT), 4 hr after treatment (*Figure 2k*), thus suggesting that NDN might stimulate thermogenesis in mice.

## Role of the EZ receptors in the anorexigenic effect of NDN

Given the aforementioned effect of central injection of NDN on the food intake behavior, we sought to determine the nature of the receptor relaying these effects. With regard to the sequence proximity between ODN and NDN, we next evaluated whether known EZ receptors could relay the anorexigenic effect of the NDN. Experiments performed in fasted mice have indicated that i.p. injections of flumazenil (10 mg/kg—$GABA_A$-R benzodiazepine binding site antagonist) (*Figure 3a*) and PK11195 (10 mg/kg—TSPO antagonist) (*Figure 3b*), were unable to blunt the anorexigenic effects of NDN, suggesting that the effect of NDN might not be relayed by classical benzodiazepines/EZ receptors. In contrast, our experiments indicated that icv injection of the cyclo$_{1-8}$[DLeu$^5$]ODN$_{(11-18)}$ (LV-1075; 100 ng), an antagonist of the EZ GPCR (*Leprince et al., 2001*), was capable of blocking the anorexigenic effect induced by the central injection of NDN (10 ng) (*Figure 3c*), suggesting that the anorexigenic effect of NDN was relayed by the uncharacterized EZ GPCR.

## Role of the MC4R in the anorexigenic effect of NDN

We also investigated the nature of the hypothalamic pathway relaying the anorexigenic effect of NDN in mice. Previous investigations performed in rodents, using the non-selective antagonist of the MC3/4 receptors, namely SHU-9119 (*Lanfray et al., 2013*), indicated that the anorexigenic effect

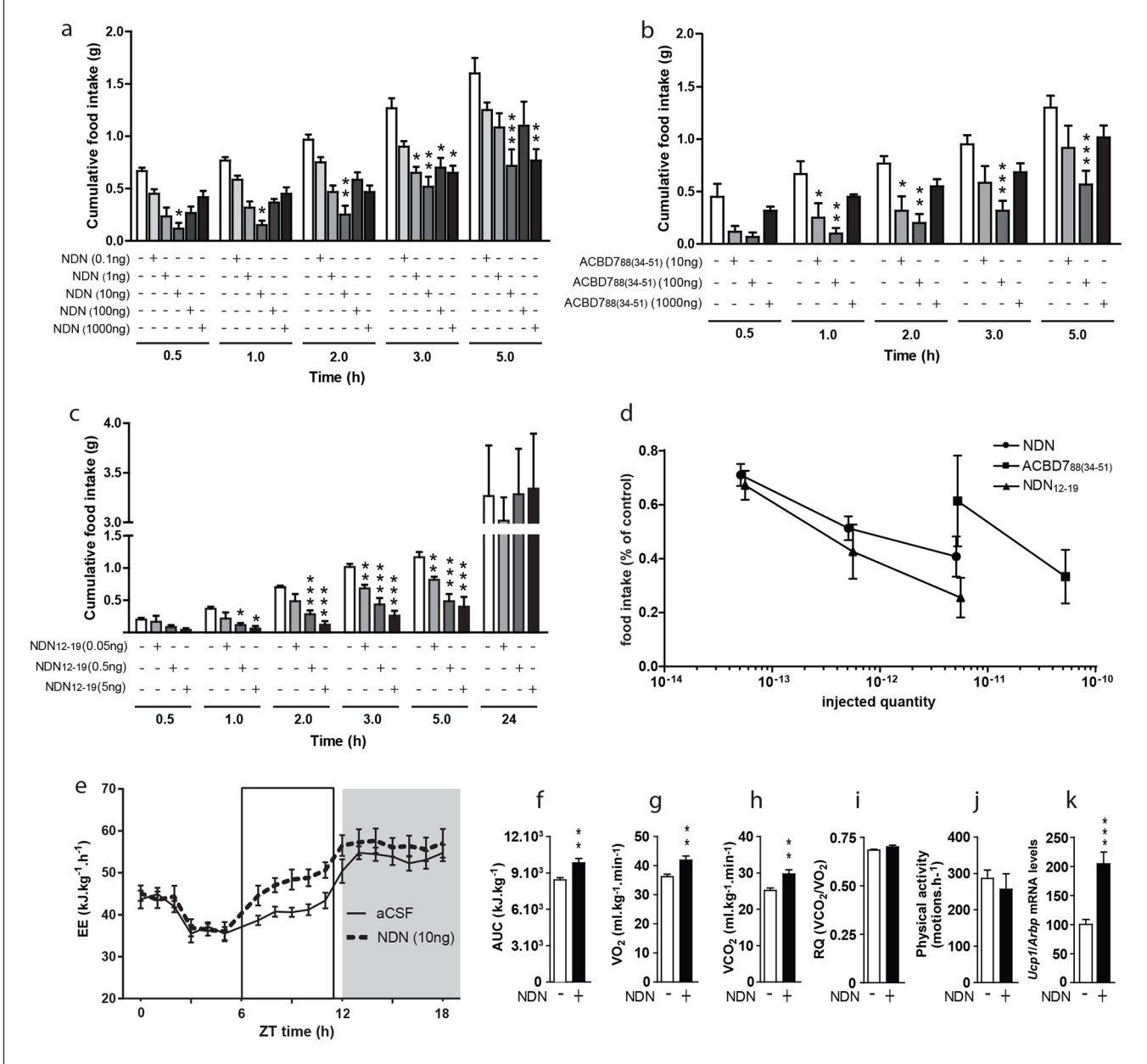

**Figure 2.** Effects of NDN on food intake and energy expenditure. (a–c) Effects of icv injection of graded doses of NDN (a), ACBD7$_{88\,(34\text{-}51)}$ (b), and NDN$_{(12\text{-}19)}$ C-terminal fragment (c) on cumulative food intake in 18 hr-fasted mice. Mice were injected in the right ventricle with the indicated substance, diluted in aCSF as vehicle, and had access to food 20 min later (*n*=5, 6). Data are expressed as mean ± SEM. Two-way ANOVA followed by a post-hoc multiple comparison Bonferroni test: *p< 0.05; **p<0.01, ***p<0.001. (d) Comparison of the efficiency of each compound in terms of inhibiting food intake 3 hr after icv injection. (e,f) Effects of icv injection of NDN (10 ng) on energy expenditure in 18 hr-fasted mice. Energy expenditure as represented by area under the curve (AUC) between 12 and 18 hr (e; grey area; *n*=8). (g–j) Effects of icv injection of NDN (10 ng) on the O$_2$ consumption (g; VO$_2$), the production of CO$_2$ (h; VCO$_2$), the respiratory quotient (i; RQ) and the locomotor activity (j) during the first 6 hr following icv injection (*n*=8). (k) iBAT Ucp1 mRNA levels 4 hr after icv injection of NDN (10 ng) (*n*=7). Data are expressed as mean ± SEM. Unpaired Student's *t* test: *p<0.05, **p<0.01, ***p<0.001.

The following source data is available for figure 2:

**Source data 1.** Impact of icv injection of NDN on food intake.

**Source data 2.** Impact of icv injection of ACBD7$_{88(34\text{-}51)}$ on food intake.

*Figure 2 continued on next page*

*Figure 2 continued*

**Source data 3.** Impact of icv injection of NDN$_{(12-19)}$ on food intake.
**Source data 4.** Impact of icv injection of NDN on energy expenditure.
**Source data 5.** Impact of icv injection of NDN on iBAT *Ucp-1* mRNA levels.

of the ODN was relayed by the activation of the melanocortin system (*Lanfray et al., 2013*). In that respect, we next evaluated the impact of NDN on the hypothalamic levels of *Pomc*, *Npy* and *Agrp* mRNAs. We observed that icv injection of NDN in fasted mice induced an increase in *Pomc* mRNA levels (*Figure 4a,b,c*) but failed to affect the mRNA levels of *Agrp* (*Figure 4d,e,f*) and *Npy* (*Figure 4g,h,i*), suggesting that NDN was able to directly activate POMC but not AgRP/NPY neurons. Converging evidence indicating that the MC4R is the prominent melanocortin receptor in the hypothalamic regulation of whole body energy homeostasis (*Butler and Cone, 2002*, *Seeley et al., 2004*, *Butler, 2006*, *Ellacott and Cone, 2006*, *Richard, 2015*), led us to evaluate the potential involvement of the MC4R signalling pathway in the effect of NDN. We observed that the anorexigenic effect induced by the icv injection of NDN (*Figure 4j*) was blocked by the injection of the MC4R-specific antagonist (HS024—100 ng). Consistently, our investigations also revealed that the icv injection of NDN failed to induce anorexigenic effects in MC4R knock-out mice (*Figure 4k*), further confirming the involvement of the melanocortin system and more specifically the MC4R as a relay in the anorexigenic effect of NDN. Additionally, investigation performed using mice hypothalamic explants revealed that NDN increases the release of the POMC-derived peptide, α-MSH, by 50% times after 45 min of treatment (*Figure 4l*). We also investigated the impact of intra-MBH injection of NDN on food intake. We observed that NDN injection in the MBH led to a significant reduction in food intake for the first 5 hr following the central administration (*Figure 4m*), suggesting that the anorexigenic effect of NDN is relayed by the activation of POMC neurons.

## Involvement of ACBD7$_{89}$ and NDN in the hypothalamic leptin-signalling pathway

A recent report indicated that *Acbd7* mRNA levels were enriched in hypothalamic cells expressing the leptin receptor (*Allison et al., 2015*). In that regard, we investigated whether NDN could contribute to the hypothalamic leptin signalling. We examined ACBD7 and NDN levels in the hypothalamus of fasted mice icv-injected with leptin (2 µg) using both Western blot and MS-MRM. We observed that the hypothalamic level of ACBD7 was increased more than 3 times (*Figure 5a and b*) while levels of NDN were increased by around 50% (*Figure 5c*) only 2 hr after the icv injection, suggesting that both ACBD7 and NDN levels are dynamically regulated by the leptin in the ARC. We observed that the anorexigenic effect induced by the icv injection of leptin was partially blunted by the co-injection of the LV-1075 (1000 ng; *Figure 5d*), while it was totally abolished by the co-injection of HS024 (100 ng; *Figure 5d*) alone or with LV-1075 (1000 ng; *Figure 5d*). Interestingly, co-injection of HS024 and LV-1075 was not associated with any additional effects on leptin induced inhibition of food intake (*Figure 5d*). Altogether, these results suggest that the endogenous EZ GPCR signalling, specifically its activation by NDN, is a part of the hypothalamic leptin signalling pathway that acts *via* the melanocortin signalling pathway. Finally, we investigated the impact of chronic injections of either NDN or LV-1075 on the food intake and body weight of mice (*Figure 5e, f*). We observed that NDN induced a transient inhibition of food intake after two days of treatment (*Figure 5e*), while a consistent reduction in body weight could be seen until the end of the treatment (*Figure 5f*). Consistently, we also observed a significant increase in both food intake (*Figure 5e*) and body weight (*Figure 5f*) in the mice injected daily with LV-1075 after two days of treatment.

## Discussion

Our results demonstrate that *Acbd7* is expressed and translated by some POMC neurons, as well as by some GABAergic neurons, but not by NPY/AgRP/GABAergic neurons in the ARC. This study also reveals that a splice variant of the *Acbd7* mRNA is expressed in the central nervous system (CNS) in

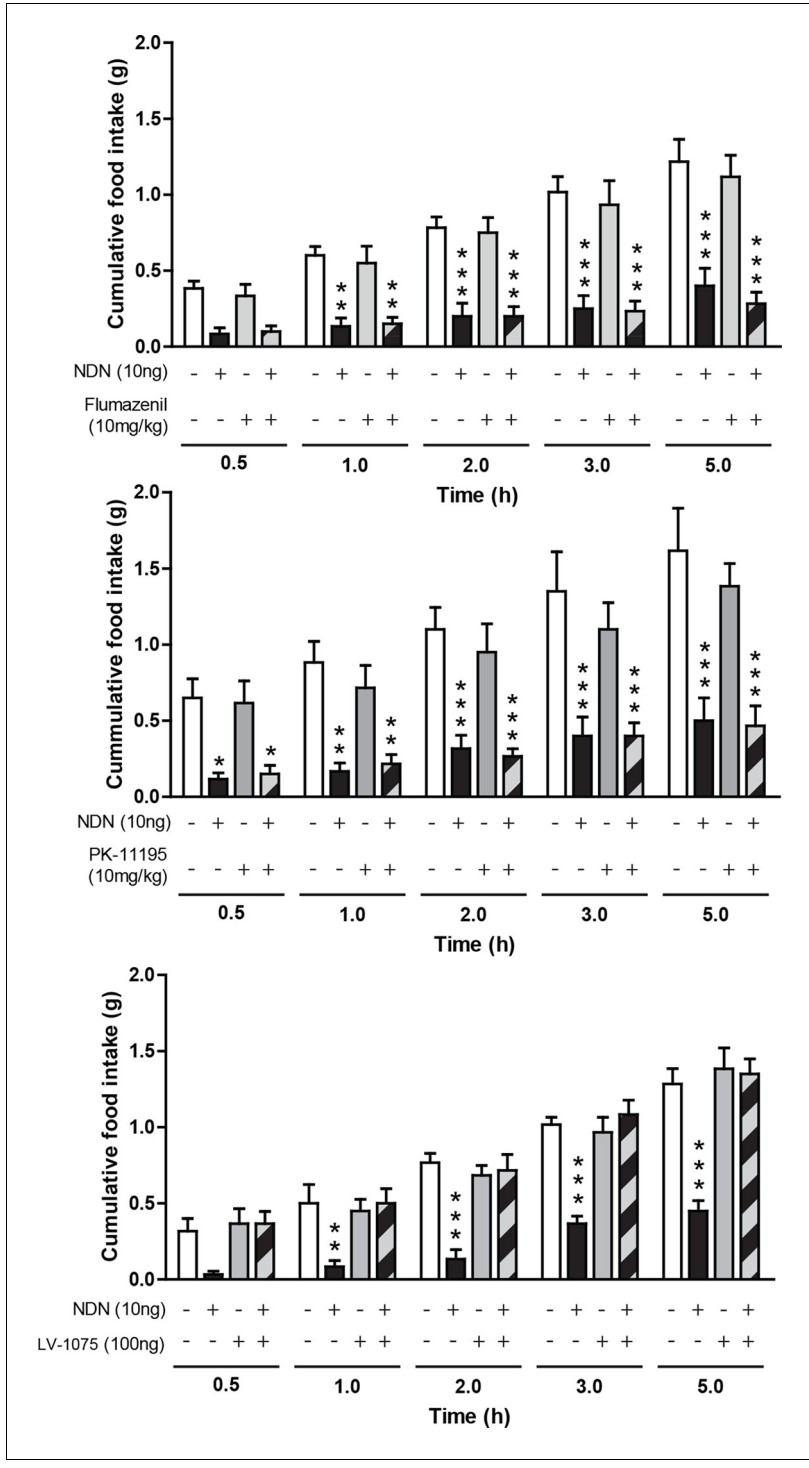

**Figure 3.** Pharmacological characterization of the receptor relaying the anorexigenic effects of NDN. Mice fasted for 18 hr received a single intraperitoneal injection of flumazenil (**a**; 10 mg/kg) or PK-11195 (**b**; 10 mg/kg) diluted in 0.9% NaCl solution, 20 min before icv injection of NDN (10 ng) or vehicle. Mice had access to food 20 min after injection and cumulative food intake was measured during the indicated periods (*n=6*). (**c**) Mice fasted for 18 hr were given icv injections of the endozepine metabotropic receptor antagonist cyclo$_{1-8}$[DLeu$^5$]ODN$_{(11-18)}$ (LV-1075; 100 ng) and NDN (10 ng) (*n=6*). Mice had access to food 20 min after icv injection and cumulative food intake was measured at the indicated periods. Data are expressed as mean ± SEM. Two-way ANOVA followed by a post-hoc multiple comparison Bonferroni test:*p< 0.05; **p<0.01, ***p<0.001.

*Figure 3 continued on next page*

*Figure 3 continued*

The following source data is available for figure 3:

**Source data 1.** Impact of flumazenil treatment on the anorexigenic effect of NDN.
**Source data 2.** Impact of PK-11195 treatment on the anorexigenic effect of NDN.
**Source data 3.** Impact of LV-1075 treatment on the anorexigenic effect of NDN.

response to catabolic signals. This splice variant encoded a protein referred as $ACBD7_{89}$ which is processed into a bioactive 19-amino-acid fragment, that we called NDN (i.e. $ACBD7_{89\ (34-52)}$). Furthermore, our experiments performed in mice indicate that NDN constitutes a anorexigenic peptide, seemingly acting through an uncharacterized EZ GPCR receptor and subsequently through a pathway involving the MC4R signalling. Finally, this work underlines the potential involvement of the ARC $ACBD7_{89}$-producing neurons, in the hypothalamic leptin signalling pathway as well as in the endogenous control of energy homeostasis.

At the time this study was initiated, the only information regarding the expression of *Acbd7* in the mouse brain consisted in microarray data provides by the Brainstars database (http://brainstars. org/probeset/1430107_at?hide=1). Here we confirm that *Acbd7* mRNA is found in selective regions in the mouse CNS. Those regions include the accumbens nucleus (Acb), median preoptic nucleus (MnPO), supraoptic nucleus (SON), PVN and ARC, which represent structures known to be involved in various behavioural and physiological regulation processes, including energy homeostasis (*Morton et al., 2006*, *Schwartz et al., 2000*, *Richard, 2015*). The distribution of *Acbd7* mRNA contrasts with that of *Dbi/Acbp* mRNA, which is widely distributed in the rodent brain (*Tonon et al., 1990*, *Rouet-Smih et al., 1992*). Our results also reveal that the unexpected splice variant isoform of *Acbd7* mRNA (i.e. *Acbd7*-002), which produces a 89 amino acid-containing protein ($ACBD7_{89}$), is expressed in the mouse brain. Interestingly, the additional glycine in the resulting $ACBD7_{89}$ protein sequence, is also present in the rat ACBD7 (http://www.ncbi.nlm.nih.gov/protein/NP_001119551.1), suggesting that this amino acid is relevant for the biological functions of NDN in rodents.

Our western blot experiments indicate that ACBD7 is produced in the mouse CNS. Furthermore, *in silico* analyses suggest that, as for the DBI/ACBP, $ACBD7_{89}$ is likely to be cleaved by tryptic digestion into shorter fragments such as NDN (*Ferrero et al., 1984*). MRM-MS experiments further indicated that endogenous NDN was found in mouse MBH lysates, confirming that hypothalamic cells are capable of processing $ACBD7_{89}$ isoform into shorter fragments. This result further underlines that maturation processes of $ACBD7_{89}$ and DBI/ACBP, which lead to the production of NDN and ODN, respectively, have been well conserved during the evolution, suggesting that both maturation products have essential physiological functions. Nonetheless, our investigation does not allow for excluding the endogenous production of the $ACBD7_{88\ (34-51)}$ fragment in the mouse CNS. Finally, given the structure homology between ACBD7 and DBI/ACBP, and specifically the conservation of the $Lys^{18}$ in the ACBD7 sequence, one can argue that brain cells could have the ability to produce several TTN-like compounds consisting of $ACBD7_{88\ (18-51)}$ and $ACBD7_{89\ (18-52)}$. Likewise, considering the marked selective pressure that have occurred on those two paralog genes, future investigations will be important in assessing the ability of ACBD7 to act as an acyl-CoA carrier, as is for DBI/ACBP.

Previous investigations have also characterized *Dbi/Acbp*, as exclusively produced by glial cells in the CNS (*Tong et al., 1990*, *Tonon et al., 1990*). However no information regarding the nature of *Acbd7*-expressing cells was available prior to our investigation. Using a double immunohistochemistry (IHC) approach, we showed that ACBD7 appeared to be exclusively produced by neuronal cells of the MBH. This result indicates that, in addition to glial cells (*Tonon et al., 1990*), neurons are also able to produce ODN-like compounds. Our study, which mainly focused on the ARC, further indicated that ACBD7 is produced by some POMC neurons, as well as by some GABAergic neurons. Our investigation also revealed that ACBD7 is not produced by NPY/AgRP/GABAergic neurons. Altogether these results suggest that ACBD7 is produced by some POMC neurons and GABAergic neurons distinct from the NPY/GABAergic neurons.

Interestingly, a previous study conducted on astrocytes reported that the release of the DBI-derived fragment ODN, which does not contain signal peptide, involved an autophagy-based

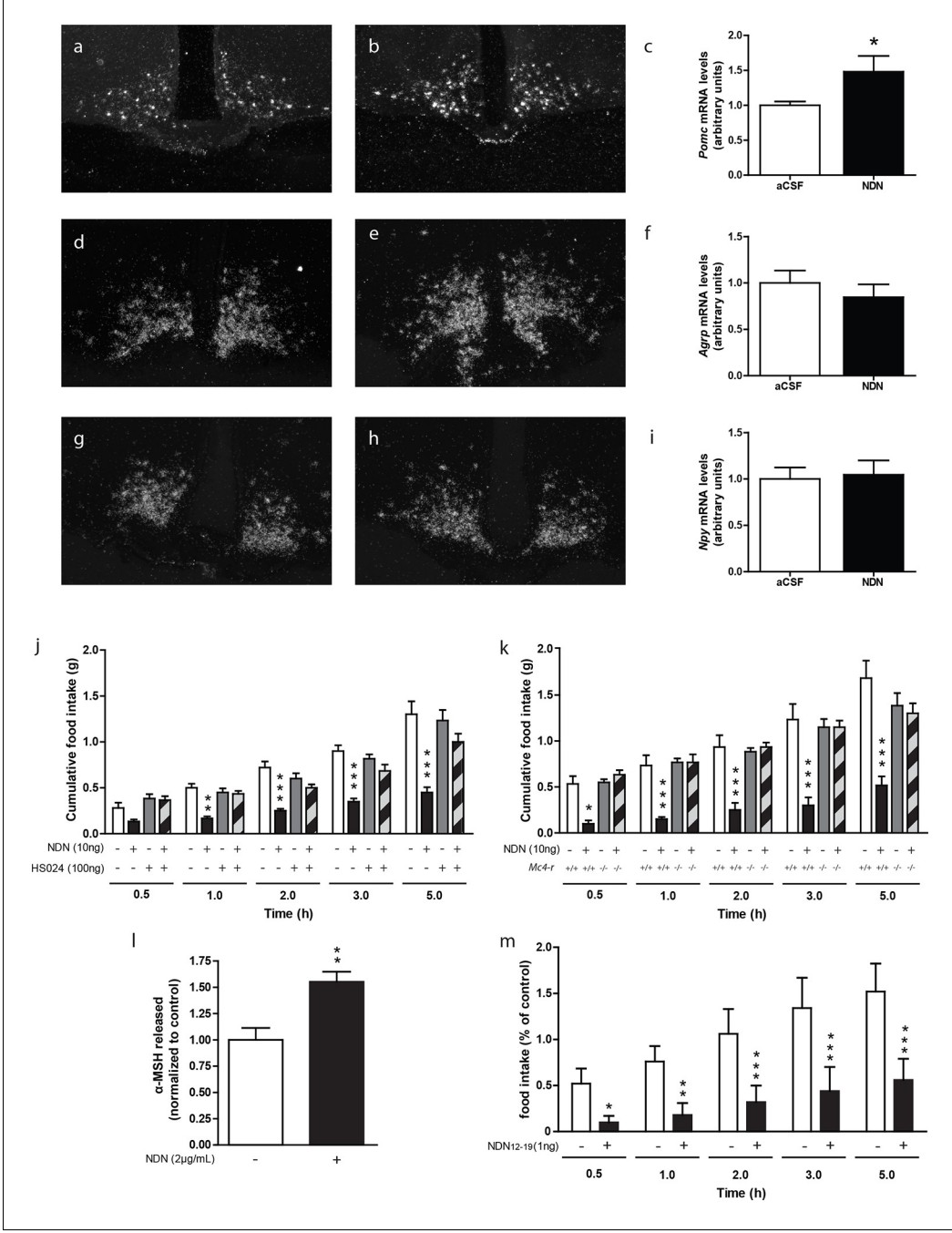

**Figure 4.** The melanocortin system relays the anorexigenic effects of NDN. (a–i) *In situ* hybridization analysis of *Pomc* mRNA levels in the mediobasal hypothalamic sections. Hypothalamic *Pomc* (a,b,c), *AgRP* (d,e,f) and *Npy* (g, h,i) mRNA levels from 18 hr-food deprived mice icv-injected with aCSF (a,d,g) or NDN (b,e,h; 100 ng) 2 hr before sacrifice. (c,f,i) Relative quantification performed in the ARC. Data were compared to aCSF injected mice as control (*n*=6, 7). Data are expressed as mean ± SEM. Unpaired Student's *t* test: *p<0.05. (j) Mice fasted for 18 hr received icv injection of an MC4R-specific antagonist (HS024; 100 ng) alone or with NDN (10 ng) diluted in aCSF (*n*=5, 6). (k) *Mc4r* knock-out or wild type mice fasted for 18 hr were icv injected with NDN (10 ng). Mice had access to food 20 min after icv injection and cumulative food intake was measured at the indicated period (*n*=6). Data are expressed as mean ± SEM. Two-way ANOVA followed by a post-hoc multiple comparison Bonferroni test: *p< 0.05, **p<0.01, ***p<0.001. (l) Hypothalamic explants were preincubated in aCSF alone followed by an incubation with or without NDN (2 μg/ml). α-MSH released during incubation was normalized to the amount released during the preincubation period (n=5). Data are expressed as mean ± SEM. Unpaired t test: *p<0.05; NS, not statistically different. (m) Mice fasted for 18 hr were bilaterally injected in the MBH with NDN (1 ng). Mice had access to food

*Figure 4 continued on next page*

*Figure 4 continued*

20 min after injection and cumulative food intake was measured at the indicated period (*n*=5). Data are expressed as mean ± SEM. Two-way ANOVA followed by a post-hoc multiple comparison Bonferroni test: *p< 0.05, **p<0.01, ***p<0.001.

The following source data is available for figure 4:

**Source data 1.** Impact of NDN on the hypothalamic *Pomc, AgRP,* and *Npy* mRNA levels.
**Source data 2.** Impact of HS024 treatment on the anorexigenic effect of NDN.
**Source data 3.** Impact of NDN on food intake in MC4R-KO mice.
**Source data 4.** Impact of NDN on $\alpha$-MSH release by hypothalamic explants.
**Source data 5.** Impact of intra-MBH injection of NDN on food intake.

unconventional secretory pathway (*Loomis et al., 2010*). In this study, 3-MA, a potent inhibitor of the autophagosome formation was shown to inhibit degradation of DBI/ACBP leading to its accumulation. In our study, we did not observe immunolabeling of ACBD7 without prior treatment with either colchicine or 3-MA. These results suggest that similar to DBI/ACBP, ACBD7 is likely processed into NDN and secreted in vivo by an autophagosome-based unconventional secretory pathway.

The expression of *Acbd7* (and the production of its NDN fragment) in the ARC provides a neuroanatomical support for a role of this gene in the hypothalamic regulation of energy balance. Further supporting this role are our data, which show that *Acbd7* mRNA levels, as well as the production of ACBD7 and NDN, correlate with the energy homeostasis status in the ARC.

To further substantiate the role of *Acdb7* in energy homeostasis, we assessed the effects NDN on food intake and energy expenditure and demonstrated the anorectic and thermogenic effects of this peptide. Moreover, our investigation reveals that NDN is more potent than ACBD7$_{88}$ $_{(34-51)}$ in reducing food intake, which suggests that the additional glycine of the ACBD7$_{89}$ sequence contributes to the biological activity of the ACBD7$_{89}$-derived peptides. Moreover, it is noteworthy that the anorexigenic effect of NDN appears to be stronger than that of ODN in mice (*de Mateos-Verchere et al., 2001*, *do Rego et al., 2007*). Interestingly, our investigations also reveal, as it has already been described for ODN (*de Mateos-Verchere et al., 2001*, *do Rego et al., 2007*, ), that the C-terminal octapeptide of NDN, also acts as potent anorexigenic peptide. Furthermore, the anorexigenic effect of NDN$_{(12-19)}$ was not observed 24 hr after icv injection, suggesting that the half-life of NDN-related peptides is short in vivo. In addition, we also observed that dose response investigations performed with either NDN or its related peptides resulted in a U-shaped dose response curve, suggesting that high concentration of NDN can non-specifically stimulate hypothalamic orexigenic pathways. Additionally, NDN induced an increase in energy expenditure in fasted mice, further supporting the role of this peptide in energy homeostasis.

Pharmacological experiments performed in mice have demonstrated that the anorexigenic effect of ODN is relayed by the activation of a GPCR, still uncharacterized distinct from the classical benzodiazepine/EZ receptors (i.e. the GABA$_A$-R and the TSPO) (*do Rego et al., 2007*). The present results indicate that the anorexigenic effect of NDN is relayed by neither the GABA$_A$-R nor the TSPO, suggesting that even though NDN could bind classical EZ receptor, the anorexigenic effect of this ACBD7$_{89}$-derived peptide seems not to be relayed by those receptors. In contrast, our investigation also revealed that the anorexigenic effect of NDN was blunted by the co-injection of the antagonist of the EZ GPCR, suggesting that similarly to ODN, NDN is also capable to act via the EZ GPCR to induce its effect. Further investigations will be necessary to fully decipher the identity of the uncharacterized EZ GPCR and to assess the ability of ACBD7-derived peptides to bind the GABA$_A$-R and the TSPO.

From the potential neuroanatomical association of ACBD7 with POMC neurons in ARC, we were interested in further studying the link between ACBD7 and the melanocortin system. Using *in situ* hybridization analysis, we demonstrate that the icv injection of NDN increases the ARC levels of

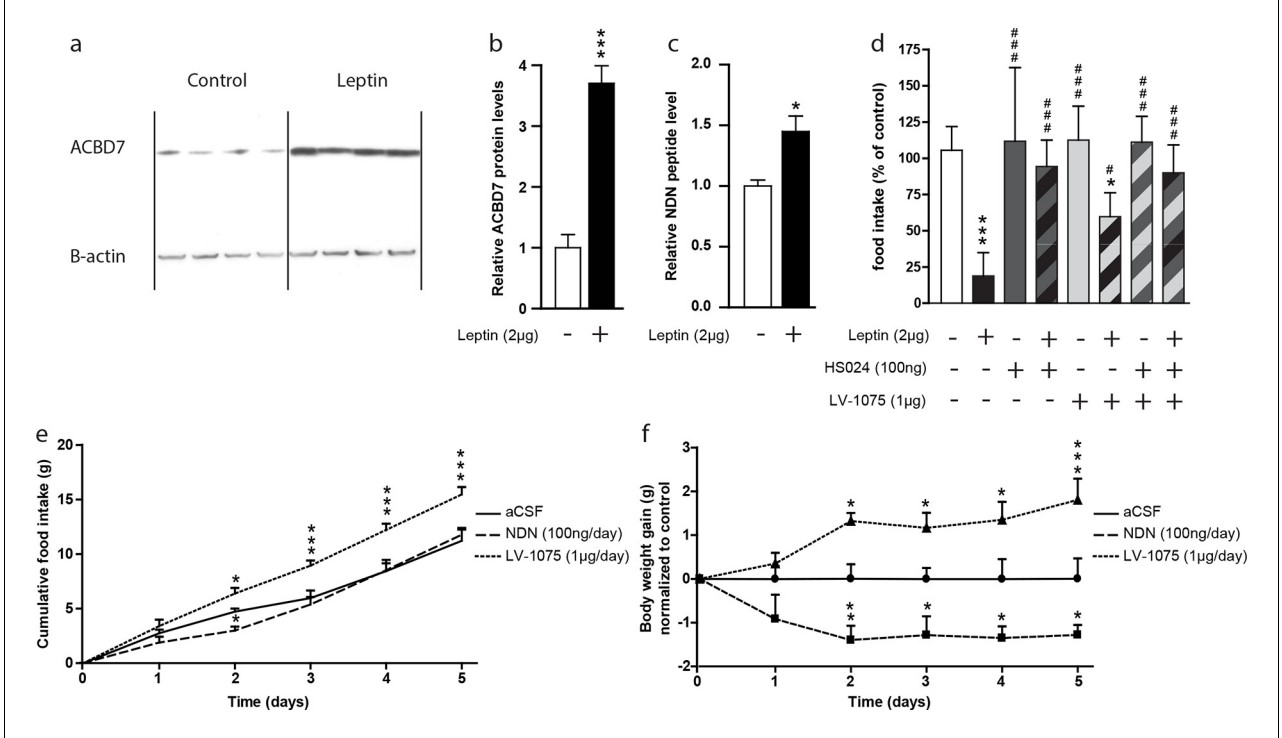

**Figure 5.** Hypothalamic NDN production is potentially involved in the leptin signaling pathway and the control of energy homeostasis. (a,b) Western blot analysis of hypothalamic protein lysates from 18 hr-fasted mice icv-injected with aCSF (control) or leptin (Leptin; 2 μg) 2 hr before sacrifice, performed using ACBD7 and β-actin specific antibodies. Quantification of relative ACBD7 protein levels performed using β-actin signal as loading control (n=4). Data are expressed as mean ± SEM. Unpaired Student's t test: ***p<0.001. (c) MRM-MS relative quantification of NDN levels in MBH explants harvested from 18 hr-fasted mice and icv-injected with aCSF, or with leptin (Leptin, 2 μg), 2 hr before sacrifice. Relative quantity was determined using exogenous peptide as an internal control (n=3). Data are expressed as mean ± SEM. Unpaired Student's t test: *p<0.05. (d) Ad libitum-fed mice received icv injection of leptin (2 μg) with or without the endozepine metabotropic receptor antagonist cyclo$_{1-8}$[DLeu$^{5}$]ODN$_{(11-18)}$ (LV-1075; 1 μg) and/or the MC4R antagonist HS024 (100 ng), 2 hr before the beginning of the dark period, and food intake was measured after 4 hr (n=6). Data are expressed as mean ± SEM. One-way ANOVA followed by a post-hoc multiple comparison Bonferroni test using aCSF injected mice as control ***p<0.001; *p<0.05, or leptin injected mice as control # p<0.05, ### p<0.001. (e,f) Effect of two daily injections of NDN (100 ng/day) and LV-0175 (1 μg/day) on food intake (e) and body weight (f). Mice were injected each day at ZT2 and ZT10. Daily food intake (e) and body weight (f) were measured each day at ZT2 for 5 days (n=6). Data are expressed as mean ± SEM. Two-way ANOVA followed by a post-hoc multiple comparison Bonferroni test: *p< 0.05; **p<0.01, ***p<0.001.

The following source data is available for figure 5:

**Source data 1.** Impact of leptin treatment on hypothalamic ACBD7 protein levels.
**Source data 2.** Impact of leptin treatment on hypothalamic NDN levels.
**Source data 3.** Impact of acute pharmacological disruption of the EZ GPCR and the MC4R signaling pathways on the anorexigenic effect of leptin.
**Source data 4.** Impact of chronic treatment (2 icv injections / day) of NDN on food intake.
**Source data 5.** Impact of chronic treatment (2 icv injections / day) of NDN on body weight.

*Pomc* mRNA without affecting *Agrp* or *Npy* mRNA levels, which suggests that ARC ACBD7$_{89}$-producing neurons may directly activate POMC neurons. Additionally, using acute pharmacological blockade of the MC4R signalling pathway as well as the MC4R knockout mouse model, we demonstrated that the anorexigenic effects of NDN were relayed via the melanocortin system. Furthermore, experiments performed on mice hypothalamic explants revealed that NDN was able to increase the secretion of α-MSH, suggesting that NDN is able to activate POMC neurons in the ARC. In addition, we demonstrated that intra-MBH injections of NDN induce strong anorexigenic

effects. Altogether, these results strongly support that ARC ACBD7$_{89}$-producing neurons are involved in the hypothalamic regulation of food intake and energy expenditure by acting on POMC neurons through releasing NDN. However, the full mechanisms whereby NDN modulates the melanocortin system tone have yet to be fully deciphered. The involvement of the melanocortin system in the action of NDN recapitulates that seen with ODN. Previous investigations performed in rodents have indeed revealed the ability of ODN to increase *Pomc* mRNA level (*Compere et al., 2003*) and the potential involvement of the MC4R receptor in the anorexigenic effect of the ODN (*Lanfray et al., 2013*).

Because of the acknowledged role of the leptin - melanocortin association in energy homeostasis (*Goncalves et al., 2014*, *Hill et al., 2010*) and the involvement of the MC4R in the effects of NDN, we hypothesized that *Acdb7* neurons could be part of the leptin-melanocortin integrated pathway. In that respect, it is noteworthy that recent data indicated that leptin effects on the melanocortin system could be relayed through an uncharacterized class of ARC GABAergic cells diverging from POMC and NPY/AgRP neurons (*Balthasar et al., 2004*, *Hill et al., 2010*, *van de Wall et al., 2008*, *Vong et al., 2011*). Our results indicate that the icv injection of leptin led to an increase in the hypothalamic levels of both ACBD7 and NDN, supporting a connection between leptin and *Acdb7* neurons. In that context, we sought to determine the involvement of the endogenous NDN production as part of the hypothalamic leptin signalling pathway. Investigations performed on *ad libitum* fed mice showed that the anorexigenic effect induced by an icv injection of leptin was partially blunted by the co-injection of the LV-1075, indicating that the hypothalamic activation of the EZ GPCR is part of the hypothalamic leptin signalling pathway. Additionally, our investigation revealed that the anorexigenic effect of icv injection of leptin is totally abolished by the co-injection of HS024, confirming that the melanocortin signalling pathway constitutes a major relay in the hypothalamic leptin signalling pathway (*Balthasar et al., 2004*). Interestingly, no additional effects were seen when LV-1075 and HS024 were co-injected, suggesting that NDN relayed the anorexigenic effect of leptin mainly through activating POMC neurons. Although, this result does not establish a direct link between NDN and leptin signalling, it nonetheless provides further evidence for such a link i.e. LV-1075 also blocked NDN anorectic action. It is noteworthy that the hypothalamic *Dbi/Acbp* mRNA level is unaffected by central injection of leptin (*Compere et al., 2010*), suggesting that glial EZs (such as ODN) are not involved in the hypothalamic leptin signalling pathway. Moreover, our observations and speculations are in line with a recent report from the translating ribosome affinity purification analysis of hypothalamic LepR positive neurons, indicating that the *Acbd7* mRNA level is enriched by more than four times in hypothalamic neurons expressing the *Lepr* (*Allison et al., 2015*).

From the potential involvement of endogenous NDN production in the hypothalamic leptin signalling pathway, we were interested in further assessing the physiological involvement of endogenous NDN signalling pathway, as well as the impact of long-term exposure to NDN, on the hypothalamic regulation of energy balance.

Interestingly, we found that NDN induced a significant reduction in body weight, which could be observed after two days of treatment, while it induced a transient inhibition of food intake that could only be seen on the second day of treatment, suggesting the involvement of compensatory mechanisms as previously shown in rodents for several anorexigenic neuropeptides including the α-MSH (*Lucas et al., 2015*) and the ODN (*de Mateos-Verchere et al., 2001*). Additionally, we observed that chronic pharmacological disruption of the EZ GPCR signalling induced a potent increase in both food intake and body weight, suggesting that the endogenous endozepines signalling pathway could be a part of the hypothalamic regulation of energy homeostasis. Although, these results do not establish a direct link between endogenous production of NDN and the hypothalamic regulation of energy homeostasis, they nonetheless provide further evidence for the involvement endozepines in this process. Additionally, it is noteworthy that single nucleotide polymorphism in both the DBI/ACBP and ACBD7 are linked with morbid obesity in humans (*Comuzzie et al., 2012*), suggesting that these two genes are complementary involved in the hypothalamic regulation of energy balance.

In conclusion, our study identifies ACBD7$_{89}$ and its maturation product NDN as potential hypothalamic controllers of energy intake and energy expenditure. Our research further indicates that NDN acts as a significant anorexigenic signal through the mediation of an uncharacterized EZs GPCR and via the melanocortin signalling pathway. This work and that from others also provides converging evidence suggesting that MBH ACBD7$_{89}$-producing neurons are part of the

hypothalamic leptin-melanocortin system as well as a major actor in the hypothalamic control of energy homeostasis.

## Material and methods

### Animals and surgical procedures

Adult male C57BL/6 mice weighting 25–30 g, and MC4R knockout mice (*Balthasar et al., 2005*, *Rossi et al., 2011*) weighting 45–50 g, were housed under constant temperature (22°C) in a 12/12h light/dark cycle with free access to standard rodent chow (Teklab lab animal diet, Montreal, Canada) and drinking water. For icv injection experiments, mice were stereotaxically implanted with a permanent 22-gauge single-guide cannula (Plastics One, Roanoke, Virginia) aimed at the lateral ventricle using the following stereotaxic coordinates: 0.4 mm posterior to the bregma, 1 mm lateral to the bregma and 2 mm ventral to the skull surface. For intra-MBH injection experiments, mice were stereotaxically implanted with a permanent 22-gauge bilateral-guide cannula (Plastics One) targeting the MBH (5.8 mm depth, 1.8 mm caudal to bregma, 0.4 mm lateral from the sagittal suture). The guide cannulas were secured with screws and cranioplastic cement (Dentsply Canada, Woodbridge, Canada, ON). To prevent clogging and to reduce the potential for brain infection, sterile obturators (Plastics One) were inserted into the guide cannulas. For icv injection, cannula placement was functionally verified before experiments using the dipsogenic effects of angiotensin 2 injections (5 ng) as positive control. For intra-MBH injection, cannula placement was confirmed *post-mortem*.

Experiments were conducted according to the Laval University Animal Ethic Committee and the Canadian Guide for the Care and Use of Laboratory Animals.

### Mouse *ACBD7* mRNA sequencing

Mouse hypothalamic mRNA was purified using RNeasy Mini Kit (Qiagen, Toronto, Canada, ON), according to the manufacturer's instructions. The RNA concentrations were estimated from absorbance at 260 nm. cDNA synthesis was performed using random hexamer primers and the expand reverse transcriptase (Roche Diagnostics, Laval, Canada, QC). The *Acbd7* cDNA was amplified using primers specific to the *Acbd7* sequence (*Acbd7*-FP: 5'-TCCGTGTCTCATCATTATGTCC-3'; *Acbd7*-RP: 5'-AGGTAACCATGCTGACAGTCCT-3') designed to cover the entire open reading frame of the *Acbd7* mRNA (410-bp). After purification on agarose gel, PRC product was sequenced by the molecular platform of Laval University (Quebec City, Canada, QC).

### *Acbd7* probe design

The mouse *Acbd7* cDNA riboprobe was prepared from a 410-bp fragment of the entire open reading frame of the *Acbd7* mRNA using the previously described primers (*Acbd7*-FP and *Acbd7*-RP). After being subcloned, into a PGEM-T plasmid (Promega, Madison, Wisconsin), and linearized with NcoI and SpeI (New England Biolabs, Whitby, Canada, ON), for antisense and sense probes respectively, radioactive riboprobe was synthesized by incubating the linearized plasmid (250 ng) in 10 mM NaCl, 10 mM 1,4-dithiothreitol, 6 mM $MgCl_2$, 40 mM Tris (pH 7.9), 0.2 mM ATP/GTP/CTP, 100 µCi $\alpha$-$^{35}$S-UTP (PerkinElmer, Foster City, California), 40 U RNase inhibitor (Roche Diagnostics), and 20 U of RNA polymerase (SP6 or T7 for antisense and sense probes, respectively) for 60 min at 37°C. The riboprobe was purified using the RNeasy Mini Kit (Qiagen), eluted in 150 µl of 10 mM Tris/1 mM EDTA buffer, and incorporated in a hybridization solution containing $10^7$ cpm of $^{35}$S probe (per ml), 52% formamide, 330 mM NaCl, 10 mM Tris, pH 8, 1 mM EDTA, pH 8, Denhardt's solution 1x, 10% dextran sulfate, 0.5 mg/ml tRNA, 10 mM 1,4-dithiothreitol, and diethyl pyrocarbonate-treated water. This solution was mixed and heated at 65°C before being spotted on slides. The specificity of the probe was confirmed by the absence of positive signal in sections hybridized with the sense probe.

### *In situ* hybridization

After having been harvested, brains were kept in paraformaldehyde (4%) for 7 days, transferred into a solution containing paraformaldehyde (4%) and sucrose (10%) and thereafter cut using a sliding microtome (Histoslide 2000, Heidelberger, Germany). Brain sections (25 µm) were taken from the olfactory bulb to the brainstem and stored at -30°C in a cryoprotective solution containing sodium phosphate buffer (50 mM), ethylene glycol (30%), and glycerol (20%). They were mounted onto

poly-L-Lysine-coated slides. The *Acbd7* mRNA as well as *Pomc, Npy* and *Agrp* mRNAs were localized by *in situ* hybridization as previously described (*Baraboi et al., 2010*). After hybridization the slides were exposed on X-ray film (Eastman Kodak, Rochester, New York) for 5 days (*Acbd7*) or for 1 day (*Pomc, Npy, Agrp*). Once removed from the autoradiography cassettes, the slides were defatted in toluene and dipped in NTB2 nuclear emulsion (Kodak, Oakville, Canada, ON). The slides were exposed for 5 weeks (*Acbd7*) or for 1 week (*Pomc, Npy, Agrp*) before being developed in D19 developer (Kodak) for 3.5 min at 14–15°C and fixed in rapid fixer (Kodak) for 5 min. Finally, tissues were rinsed in running distilled water for 2–3 hr, counterstained with thionin (0.25%), dehydrated through graded concentrations of alcohol, cleared in toluene, and coverslipped with dibutylphtalate-xylol (DPX) mounting medium.

## *In silico* analysis of ACBD7 potential processing

Analysis of amino acid conservation between mouse DBI/ACBP and ACBD7 was performed using the DIALIGN software (http://bibiserv.techfak.uni-bielefeld.de/dialign/) and using the ACBD7 described sequence available from the NCBI (http://www.ncbi.nlm.nih.gov/protein/NP_084339.1) and Ensembl website (http://useast.ensembl.org/Mus_musculus/Gene/Family?family=ENSFM00670001245867;g=ENSMUSG00000026644;r=2:3336168-3340993;t=ENSMUST00000115089) and the ACBD7 splice variant expected sequence as template. Prediction of potential cleavage sites in the $ACBD7_{88}$ and $ACBD7_{89}$ protein was performed by using the Vector NTI advance 11 software (Invitrogen, Burlington, Canada, ON).

## Western blot

Antibody directed against ACBD7 was obtained from Novus Biologicals (Oakville, Canada, ON; NBP1-56527). Rabbit secondary antibody was purchased from Santa Cruz Biotechnology (Santa Cruz Biotechnology, Santa Cruz, California). Tissues (10 mg) were homogenized in a lysis buffer composed of 50 mM HEPES (pH 7.4), 40 mM NaCl, 2 mM EDTA, 10 mM sodium pyrophosphate (Sigma, Oakville, Canada, ON), 10 mM sodium glycerophosphate (Sigma), 50 mM sodium fluoride (Sigma), and 2 mM sodium orthovanadate (Sigma), supplemented with 0.1% of sodium dodecyl sulfate, 1% of sodium deoxycholate (Sigma), and 1% of Nonidet-P40 (Sigma). One tablet of protease inhibitor cocktail (Roche, Indianapolis, United Stated, IN) and one tablet of phosphatase inhibitor cocktail (Roche) were added per 10 ml of lysis buffer. Tissues were rotated at 4°C for 10 min, and then the soluble fractions were isolated by centrifugation at 13,000 rpm for 10 min in a refrigerated microcentrifuge. Protein levels were then quantified using a protein assay dye reagent concentrate (Bio-Rad, Mississauga, Canada, ON), and analyzed by western blotting. Briefly, 20 µg of proteins were electrophoresed (2 hr, 110 V) onto NuPAGE Novex 8–16% Bis-Tris gels (Invitrogen, Burlington, Canada, ON) using TRIS-Glycine running buffer. Proteins were transferred onto polyvinylidene fluoride (PVDF) membrane (Bio-Rad) using the Trans-Blot Turbo transfer system (Bio-Rad) in transfer buffer for 7 min (1.3 A; 25V). After the transfer, the PVDF membrane was washed in phosphate-buffered saline (PBS)-Tween and incubated in blocking buffer (5% non-fat milk in PBS-Tween) for 1 hr at room temperature. The membrane was then washed and incubated overnight at 4°C with antibody directed against ACBD7 (1:1,000 in 5% non-fat milk in PBS-Tween). The membrane was washed and incubated for 1 hr at room temperature with the secondary antibody (1:10,000 in 5% non-fat milk in PBS-Tween). After washing four times, the membrane was incubated with chemiluminescent substrate, ECL (Amersham Biosciences, Arlington Heights, United Stated, IL), for 1 min, and signal was detected by exposing the membranes on a ECL-hyperfilm.

### Immuhistochemistry experiments

Mice were centrally injected with colchicine (Tocris, Minneapolis, Minnesota - 100 ng) or 3-methyladenin (Tocris - 3-MA, 50pmol), 24 hr before the sacrifice. Brains were perfused with Bouin fixative solution, harvested and embedded in paraffin. Mice brain coronal sections (7µm) were sequentially incubated at room temperature in methanol containing 3% of $H_2O_2$ (30 min), PBS containing 0.1% of triton X-100 (10 min), HCl (1N, 2 min) and PBS containing 1% of normal goat serum (NGS) for 1 hr. They were then incubated at 4°C for 18 hr with rabbit affinity-purified antiserum directed again ACBD7 (1:400; Novus biologicals). Sections were rinsed with PBS, incubated with SignalStain Boost IHC detection reagent (Cell Signaling, Whitby, Canada, ON; 30 min, RT) and labelled using

SignalStain DAB substrate kit (Cell Signaling). After inactivation of the initial reaction with methanol containing 3% of $H_2O_2$ (20 min, RT), secondary immunolabelling was processed. Coronal sections were sequentially incubated in PBS containing 1% of NGS and 0.1% Triton X-100 for 1 hr, then slices were incubated at 4°C for 18 hr with rabbit purified antiserum directed again the GFAP (DAKO, 1/400), the POMC (Phoenix Pharmaceuticals, Burlingame, California, 1/500), the NPY (Phoenix Pharmaceuticals, 1/500) or the VGAT (Novus biologicals, 1/400). After having been rinsed with PBS, coronal sections were incubated with DAB/Nickel peroxidase substrate (Vector Laboratories, Burlington, Canada, ON). After washes in PBS, slices were mounted in DPX.

## Proteomic analysis of ACBD7 in vivo processing

Ability of hypothalamic cell to process ACBD7 in vivo has been evaluated using multiple reactions monitoring mass spectrometry (MRM-MS) analysis of hypothalamic explant (Proteomic Centre Facility of the Laval University, Québec City, Canada, QC). Briefly, fresh MBH hypothalamic explants were solubilized in extraction buffer containing 0.5% of sodium deoxycholate, 50 mM of 1,4-dithiothreitol and protease inhibitor. After sonication, half of the volume was precipitated by incubating with acetone (-20°C, 12 hr) and the remaining half part was purified using Amicon cartridge (cut-off 10 KDa). Both fractions were then combined and purified using HLB Oasis cartridge (Waters, Mississauga, Canada, ON). After adjustment of the sample concentration with formic acid solution (0.1%), 100 fmol/µl of exogenous synthetic peptide was added to each sample in order to normalize each analysis.

## Real time PCR assessment of *Ucp1* mRNA levels in the iBAT

Total mRNA was isolated from iBAT using QIAzol and the RNeasy Lipid Tissue Kit (QIAGEN, Mississauga, ON, Canada). The RNA concentrations were estimated from absorbance at 260 nm and cDNA synthesis was performed using the expand reverse transcriptase (Invitrogen, Burlington, ON, Canada) on 1 µg of total mRNA. mRNA extraction and cDNA synthesis were performed according to the manufacturer's instructions, and cDNA was diluted in DNAse-free water (1:30) before quantification by real-time PCR. *Ucp1* and acidic ribosomal phosphoprotein P0 (*Arbp*) mRNA levels were measured in duplicate samples using a CFX96 touch real-time PCR (Bio-Rad Laboratories, Mississauga, Canada, ON) by using primers specific to the murine *Ucp1* mRNA (*Ucp1*-FP: 3'-GCAGTG TTCATTGGGCAGCC 5'; *Ucp1*-RP: 3'- GGACATCGCACAGCTTGGTAC-5') and to the *Arbp* mRNA (Arbp-FP: 3'- AGAAACTGCTGCCTCACATC-5'; *Arbp*-RP: 3'- CATCACTCAGAATTTCAATGG-5'). Chemical detection of the PCR products was achieved with SYBR Green I (Sigma-Aldrich, Oakville, Canada, ON). At the end of each run, melt curve analyses were performed, and representative samples of each experimental group were run on agarose gel to ensure the specificity of the amplification. Fold differences in target mRNA expression were measured using the ΔΔ-cycle threshold method by comparison with the house-keeping gene and expressed as fold change between vehicle versus NDN icv-injected mice.

## Drug injections

Icv injections were performed in a final volume of 2 µL of artificial cerebrospinal fluid (aCSF; Harvard Apparatus, Saint-Laurent, Canada, QC). Synthetic $ACBD7_{88}$ $_{(34-51)}$ (H-Gln-Ser-Val-Ile-Gly-Asp-Ile-Asn-Ile-Ala-Cys-Pro-Ala-Met-Leu-Asp-Leu-Lys-OH; Thermo Scientific Pierce Protein Research, Thermo-Fisher, Waltham, Massachusetts) fragment was injected at doses between 10 and 1000 ng. Synthetic NDN (H-Gln-Ser-Val-Ile-Gly-Asp-Ile-Asn-Ile-Gly-Ala-Cys-Pro-Ala-Met-Leu-Asp-Leu-Lys-OH; Thermo-Fisher) corresponding to the $ACBD7_{89}$ $_{(34-52)}$ fragment was injected at doses between 0.1 and 1000 ng for dose-response evaluation and 10 ng for the other experiments. Synthetic $NDN_{(12-19)}$ (H-Cys-Pro-Ala-Met-Leu-Asp-Leu-Lys-OH; PRIMACEN, Rouen, France) corresponding to the $ACBD7_{89}$ $_{(45-52)}$ was injected at doses between 5 to 500 ng. The doses of icv injected $cyclo_{1-8}[DLeu^5]ODN_{(11-18)}$ (Arg-Pro-Gly-Leu-DLeu-Asp-Leu-Lys; LV-1075), HS024 and leptin were selected according to previous feeding experiments performed in mice (*do Rego et al., 2007*). Intra-MBH dual injection was performed in a final volume of 0.5 µL of aCSF (Harvard Apparatus) by cannula. Synthetic NDN (ThermoFisher) was injected at the dose of 0.5 ng by side. Intraperitoneal injections of PK11195 (10mg/Kg of body weight, Tocris) and Flumazenil (10mg/Kg of body weight, Sigma Aldrich) were performed in saline solution (NaCl, 0.9%) containing 10% DMSO, 20 min before icv injection.

## Food intake experiments

Fasted mice (ZT9 to ZT27, total 18 hr) had access to a weighed food pellet (25 g) 20 min after icv injection. Cumulative food intake was measured by briefly (less than 20 s) removing and weighing the pellet at the indicated time points as previously described (*Lanfray et al., 2013*, *do Rego et al., 2007*).

## Culture of mice hypothalamic explants

Mice hypothalamus were equilibrated in aCSF (Harvard Apparatus) at 37°C for 1 hr under constant bubbling of 95% $O_2$ and 5% $CO_2$. The explants were preincubated with fresh aCSF for 45 min followed by another incubation of 45 min with or without NDN (2 µg/mL). Measurement of α-MSH-like immunoreactivity in media was performed using ELISA kit (MyBioSource, San Diego, CA).

## Indirect calorimetry

After 48 hr of acclimation in metabolic cages, mice were fasted for 24 hr (ZT12 to ZT36). They were then icv injected at ZT30 with NDN (10 ng) or vehicle solution (aCSF), and individually monitored by indirect calorimetry to analyse $O_2$ consumption, $CO_2$ production and locomotor activity. Measurements were made continuously over 6 hr (ZT30 to ZT36) in an open circuit system with an oxygen analyzer (Applied Electrochemistry, Pittsburgh, Pennsylvania, S-3A1) and carbon dioxide analyzer (Applied Electrochemistry, CD-3A) as previously described (*Sell et al., 2004*).

## Statistical analysis

All data are expressed as mean ± SEM. Statistical analysis was performed using Student's t test or the one way ANOVA, followed by a post hoc multiple comparisons Bonferroni test (Prism 6 software, GraphPad, La Jolla, California). For each test, a value of $p < 0.05$ was considered statistically significant.

# Acknowledgements

We thank our colleagues, particularly Julie Plamondon and Pierre Samson for their technical assistance.

# Additional information

### Funding

| Funder | Grant reference number | Author |
| --- | --- | --- |
| Canadian Institutes of Health Research | Training Program in Obesity/Healthy Body Weight Research | Alexandre Caron |
| Canadian Diabetes Association | Post-doctoral fellowship | Alexandre Caron |
| Foundation of the Quebec Heart and Lung Institute Research Centre | Obesity axis fund 2013 | Denis Richard |
| Canadian Institutes of Health Research | Operating Grant MOP-142361 | Denis Richard |
| Canadian Institutes of Health Research | Operating Grant MOP-119436 | Denis Richard |
| Natural Sciences and Engineering Research Council of Canada | RGPIN-2014-06721 | Denis Richard |

The funders had no role in study design, data collection and interpretation, or the decision to submit the work for publication.

## Author contributions

DL, Conception and design, Acquisition of data, Analysis and interpretation of data, Drafting or revising the article, Contributed unpublished essential data or reagents; AC, M-CR, Acquisition of data, Analysis and interpretation of data, Drafting or revising the article; ML, FM, JL, M-CT, Analysis and interpretation of data, Drafting or revising the article, Contributed unpublished essential data or reagents; DR, Conception and design, Analysis and interpretation of data, Drafting or revising the article, Contributed unpublished essential data or reagents

## Author ORCIDs

Damien Lanfray, http://orcid.org/0000-0002-3326-9896
Alexandre Caron, http://orcid.org/0000-0001-6939-6136

## Ethics

Animal experimentation: This study was performed in strict accordance with the recommendations in Canadian Guide for the Care and Use of Laboratory Animals. The protocol was approved by the Animal Ethic Committee (CPAUL) of the Laval University (Permit Number: #2013-019-3). All surgery was performed under Isoflurane anesthesia, and every effort was made to minimize suffering.

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
