## [Decision Letter]

Thank you for submitting your work entitled "Involvement of ACBD7 in the control of food intake and energy expenditure in mice" for consideration by *eLife*. Your article has been favorably evaluated by a Senior editor and three reviewers, one of whom, Richard Palmiter, is a member of our Board of Reviewing Editors.

The reviewers have discussed the reviews with one another and the Reviewing editor has drafted this decision to help you prepare a revised submission.

Summary:

The authors have provided compelling evidence that a novel peptide derived from the *Acbd7* gene has potent inhibitory effects on food intake. The peptide appears to be produced in the novel class of neurons in the arcuate nucleus and acts on POMC neurons.

Essential revisions:

Three reviewers find the results describing the effects of this novel peptide on food intake as provocative and suitable for publication after attending to the following major issues that were raised.

1) The authors should provide good evidence that exogenous NDN peptide can dramatically affect food intake; however, more compelling evidence that NDN peptide is necessary for regulation of food intake is necessary.

2) More data related to the duration of NDN effects on food intake and the consequences of repeated injections on body weight are requested.

3) The authors should examine the possibility that ACBD7 is co-expressed in other (non-POMC, non-AgRP) populations of ARC neurons that have been described in recent literature.

4) Although, the GPCR through which NDN acts is unknown, electrophysiological experiments showing a direct action on of NDN on POMC neurons would provide important mechanistic information; authors should include such experiments if possible. The authors should also do their best to respond to the other comments raised by the reviewers that will improve the manuscript.

*Reviewer #1:*

The authors of this paper show that the *Acbd7* gene is differentially spliced in the hypothalamus to give rise to a 19-amino acid peptide (NDN) instead of the typical 18 amino acid peptide (ODN). The precursor protein is made by POMC neurons but not by AGRP/NPY neurons; its expression is reduced by fasting and induced by feeding. The peptide, results in pronounced anorexia when injected into the brain (icv) that depends on melanocortin signaling. Blockade of either the receptor for NDN or the melanocortin MC4 receptor blocks the anorectic effect of NDN. The results clearly establish the anorectic effect of NDN and begin to establish the neurons involved; however, there are several issues that need to be resolved.

1) The data establish that NDN is sufficient to reduce food intake; however, it is equally important to know whether it is necessary. The only experiment that addresses that issue is Figure 5, in which the authors show that the EZ GPCR antagonist (LV-1075) blunts the anorexic effect of leptin. The problem with this experiment is that leptin affects many different neuronal populations; thus, it is hard to draw firm conclusions about pathways. Comparing the effectiveness of LV-1075 with the MC4R antagonist (HS024) and the combination of the two would help establish the contribution of the melanocortin pathway to the anorexic effect. Another approach would be to see if LV-1075 would block a 5HT2c agonist-induced anorexia.

2) The paper presents several sets of dose-response histograms with various peptides. Many of the dose-response relationships appear to be "U" shaped – i.e. higher doses give less anorexia than middle doses. A comment on this phenomenon would be useful.

3) Also, because the peptides are of different molecular weights, it would be useful to show at least the descending portions of the dose-response curves in one graph and express the data in terms of picomoles peptide injected rather than micrograms. That way, the relative potency of the various peptides will become more obvious.

4) *Acbd7* protein does not appear to have a signal peptide; thus, it is worth discussing how the NDN is secreted from the cells that make it.

*Reviewer #2:*

In the manuscript by Lanfray et al., the authors describe the identification of novel peptide product that has potential influences on food intake and energy homeostasis in mice. Specifically, they identify a nonadeca peptide (NDN) that is produced out of the ACBD7 gene product, which is regulated in its mRNA and protein expression depending on energy status. in vivo injection studies of the NDN reduces appetite and increases energy expenditure and suggesting this is mediated via interaction with the melanocortin and leptin system.

This is a comprehensive and well-designed study which identifies a potential new player in the complex regulatory circuitries that control energy homeostasis. However, it falls a bit short on providing detailed mechanistic insights.

Comments:

One interesting aspect of this new protein is the expression pattern which seems to be different from the classical players POMC and NPY/AGRP specifically in the Arc where there seems to be little or no overlap. It would be nice to see some attempts to more closely identifying these novel type of neurons e.g. are they GABAergic? Any other neuropeptide markers that are colocalised in these neurons, etc.?

While the inhibitory effect on food intake is sticking, it is only shown for the duration of 5 hours. How long does the effect last? Are they back to normal after 24 hours?

What are the chronic effects?

Is the mRNA expression of ACBD7 only altered in the ARC in response to food depravation/refeeding or are other nuclei (e.g. PVN) also affected?

Has IHC being performed on brain slices without colchicine injection e.g. can the peptide be seen in fibres and does this give any information to which other areas these 'novel' neurons projecting too?

While I am not demanding to perform this experiments ultimate proof of the functionality as a novel regulator of energy homeostasis will have to include some chronic models e.g. transgenics/KO's.

*Reviewer #3:*

In their current contribution Lanfray et al. investigate the role of ACBD7 on energy homeostasis and feeding. They reveal transcriptional and translation control of ACBD7 in the hypothalamus dependening on feeding state, with a pronounced upregulation upon refeeding. ACBD7 appears to be primarily expressed in the ARC and PVN. They also demonstrate, that ACBD7 is cleaved to a NDN peptide, and that also NDN increases upon refeeding. Functionally, icv injection of NDN suppresses refeeding, increases energy expenditure and BAT UCP1-expression. The anorexigenic effect of icv applied NDN relies on a yet uncharacterized GPCR, but not on GABA-R-signaling. Finally, the anorexigenic effect of NDN is abrogated upon application of an MC4R-antagonist and leptin increases ACBD7-expression.

Overall, the experiments appear well performed and support the conclusions drawn by the authors. The findings are novel and interesting. However, a few parts of the study present interesting findings, which are correlative, but not fully causally linked. Important questions, which should be addressed include.

1) What is the exact quantitative proportion of POMC-cells expressing ACBD7?

2) Does NDN directly excite POMC-neurons. This should be addressed through investigating the effect of NDN on firing properties of identified POMC-neurons.

3) Given that ACBD7 appears to be regulated by leptin, it should be addressed to which extend NDN-action contributes to leptin's anorexigenic action, thus linking both aspects of the study. Here, the effect of blocking the NDN-activated GPCR on leptin's ability to suppress feeding and to evoke neuroactivation in the PVN (via c_Fos staining) should be addressed.

---

## [Author Response]

Essential revisions:

*Three reviewers find the results describing the effects of this novel peptide on food intake as provocative and suitable for publication after attending to the following major issues that were raised.*

*1) The authors should provide good evidence that exogenous NDN peptide can dramatically affect food intake; however, more compelling evidence that NDN peptide is necessary for regulation of food intake is necessary.*

In order to further demonstrate the role of NDN on energy balance we have added data demonstrating that repeated icv injections of NDN reduce body weight gain (see response to point # 2, below). Additionally, we carried out new experiments (Figure 5 and Figure 5—figure supplement 5) to examine the impact of repeated injections of LV-1075, an antagonist of the uncharacterized endozepine metabotropic receptor, also an apparent receptor of NDN. Our results demonstrate that acute pharmacological disruption of the EZ-R signaling pathway leads to an increase in both food intake and body weight during 5 days of the treatment.

*2) More data related to the duration of NDN effects on food intake and the consequences of repeated injections on body weight are requested.*

Our results indicate that the hypophagic effect of a single injection of NDN12-19 is no longer present 24 hours after a single icv injection. Moreover, we have added new data, which show that repeated icv injections NDN over 5 days led to a consistent reduction in body weight while transiently reducing food intake. These results have been added in Figure 2 and Figure 5 and have also been discussed (Discussion).

*3) The authors should examine the possibility that ACBD7 is co-expressed in other (non-POMC, non-AgRP) populations of ARC neurons that have been described in recent literature.*

To address this major point, we have performed additional immunohistochemistry experiments, which reveal that ACBD7 is colocalized with the vesicular GABA transporter (VGAT) in the ARC. This result indicates that ACBD7 is produced by both GABAergic and POMC neurons in the ARC. We have added these findings in Figure 1 and elaborated upon our findings in the Discussion section.

*4) Although, the GPCR through which NDN acts is unknown, electrophysiological experiments showing a direct action on of NDN on POMC neurons would provide important mechanistic information; authors should include such experiments if possible. The authors should also do their best to respond to the other comments raised by the reviewers that will improve the manuscript.*

Electrophysiological experiments could not be done. We nonetheless have added new data demonstrating that NDN induces MSH release from hypothalamic explants and decreases food intake after its injection into the mediobasal hypothalamus. We have added these results in Figure 4, and elaborated upon them in the Discussion section.

*Reviewer #1: The authors of this paper show that the Acbd7 gene is differentially spliced in the hypothalamus to give rise to a 19-amino acid peptide (NDN) instead of the typical 18 amino acid peptide (ODN). The precursor protein is made by POMC neurons but not by AGRP/NPY neurons; its expression is reduced by fasting and induced by feeding. The peptide, results in pronounced anorexia when injected into the brain (icv) that depends on melanocortin signaling. Blockade of either the receptor for NDN or the melanocortin MC4 receptor blocks the anorectic effect of NDN. The results clearly establish the anorectic effect of NDN and begin to establish the neurons involved; however, there are several issues that need to be resolved.*

*1) The data establish that NDN is sufficient to reduce food intake; however, it is equally important to know whether it is necessary. The only experiment that addresses that issue is Figure 5, in which the authors show that the EZ GPCR antagonist (LV-1075) blunts the anorexic effect of leptin. The problem with this experiment is that leptin affects many different neuronal populations; thus, it is hard to draw firm conclusions about pathways. Comparing the effectiveness of LV-1075 with the MC4R antagonist (HS024) and the combination of the two would help establish the contribution of the melanocortin pathway to the anorexic effect. Another approach would be to see if LV-1075 would block a 5HT2c agonist-induced anorexia.*

We sincerely thank the reviewer for his/her comments and suggestions for revision.

We have compared the effectiveness of LV-1075 and HS024 in blocking the anorexigenic effect of a central injection of leptin. Our results indicate that while HS024 almost completely blocked the anorexigenic effect of a central injection of leptin, LV- 1075 only partly (50%) reduced the anorexigenic effect. We have included these results in Figure 5, and added comments in the Discussion section.

*2) The paper presents several sets of dose-response histograms with various peptides. Many of the dose-response relationships appear to be "U" shaped* – *i.e. higher doses give less anorexia than middle doses. A comment on this phenomenon would be useful.*

We agree that this point needs clarification. We have added a new paragraph in the Discussion section to suggest that high concentration of NDN could non-specifically stimulate hypothalamic orexigenic pathways.

*3) Also, because the peptides are of different molecular weights, it would be useful to show at least the descending portions of the dose-response curves in one graph and express the data in terms of picomoles peptide injected rather than micrograms. That way, the relative potency of the various peptides will become more obvious.*

We thank the reviewer for this suggestion. We have incorporated this suggestion in Figure 2.

*4) Acbd7 protein does not appear to have a signal peptide; thus, it is worth discussing how the NDN is secreted from the cells that make it.* We thank the reviewer for this suggestion. We have added a comment in Discussion, which suggests a potential involvement of an unconventional secretory pathway in the release of NDN, as it has been proposed for the DBI/ACBP-derived peptide ODN.

*Reviewer #2: In the manuscript by Lanfray et al., the authors describe the identification of novel peptide product that has potential influences on food intake and energy homeostasis in mice. Specifically, they identify a nonadeca peptide (NDN) that is produced out of the ACBD7 gene product, which is regulated in its mRNA and protein expression depending on energy status.* in vivo *injection studies of the NDN reduces appetite and increases energy expenditure and suggesting this is mediated via interaction with the melanocortin and leptin system. This is a comprehensive and well-designed study which identifies a potential new player in the complex regulatory circuitries that control energy homeostasis. However, it falls a bit short on providing detailed mechanistic insights. Comments: One interesting aspect of this new protein is the expression pattern which seems to be different from the classical players POMC and NPY/AGRP specifically in the Arc where there seems to be little or no overlap. It would be nice to see some attempts to more closely identifying these novel type of neurons e.g. are they GABAergic? Any other neuropeptide markers that are colocalised in these neurons, etc.?*

We thank the reviewer for bringing this concern to our attention. We performed additional immunohistochemistry experiments, which reveal that ACBD7 is partially colocalized with the vesicular GABA transporter (VGAT) in the ARC. We have added these results in Figure 1 and have also added a comment in the Discussion section.

*While the inhibitory effect on food intake is sticking, it is only shown for the duration of 5 hours. How long does the effect last? Are they back to normal after 24 hours?*

We have assessed the food consumption 24 h after the acute icv injection of NDN12-19. Our results indicate that the anorexigenic effect is lost 24h after a single injection. We have added these results in Figure 2.

*What are the chronic effects?*

In response to this comment, we have investigated the impact of repeated (5 days) icv injections of NDN on both food intake and body weight. Our results demonstrate the ability of repeated injections of NDN in reducing food intake and body weight (Figure 5 and Figures 5—figure supplement 4 and 5).

*Is the mRNA expression of ACBD7 only altered in the ARC in response to food depravation/refeeding or are other nuclei (e.g. PVN) also affected?*

We have assessed the *Acbd7* mRNA levels in the PVN following fasting and refeeding challenges. Our results indicate that the mRNA levels are unaffected in the PVN. We have added these results in Figure 2, and have elaborated upon them in the Discussion section.

*Has IHC being performed on brain slices without colchicine injection e.g. can the peptide be seen in fibres and does this give any information to which other areas these 'novel' neurons projecting too?*

Immunolabeling failed to reveal projections even without prior treatment with colchicine. We have added immunolabeling images in Figure 1—figure supplement 3.

*While I am not demanding to perform this experiments ultimate proof of the functionality as a novel regulator of energy homeostasis will have to include some chronic models e.g. transgenics/KO's. Reviewer #3: In their current contribution Lanfray et al. investigate the role of ACBD7 on energy homeostasis and feeding. They reveal transcriptional and translation control of ACBD7 in the hypothalamus dependening on feeding state, with a pronounced upregulation upon refeeding. ACBD7 appears to be primarily expressed in the ARC and PVN. They also demonstrate, that ACBD7 is cleaved to a NDN peptide, and that also NDN increases upon refeeding. Functionally, icv injection of NDN suppresses refeeding, increases energy expenditure and BAT UCP1-expression. The anorexigenic effect of icv applied NDN relies on a yet uncharacterized GPCR, but not on GABA-R-signaling. Finally, the anorexigenic effect of NDN is abrogated upon application of an MC4R-antagonist and leptin increases ACBD7-expression. Overall, the experiments appear well performed and support the conclusions drawn by the authors. The findings are novel and interesting. However, a few parts of the study present interesting findings, which are correlative, but not fully causally linked. Important questions, which should be addressed include. 1) What is the exact quantitative proportion of POMC-cells expressing ACBD7?*

We agree with the reviewer that this information will be relevant. However, our immunohistological analyses is limited in terms of the specific quantification of ACBD7-labelled cells. We therefore prefer not to include an approximate value of these cells in our manuscript.

*2) Does NDN directly excite POMC-neurons. This should be addressed through investigating the effect of NDN on firing properties of identified POMC-neurons.*

Unfortunately, we cannot perform electrophysiological experiments. Please see our comments in response to the point number 4 of the essential revision above.

*3) Given that ACBD7 appears to be regulated by leptin, it should be addressed to which extend NDN-action contributes to leptin's anorexigenic action, thus linking both aspects of the study. Here, the effect of blocking the NDN-activated GPCR on leptin's ability to suppress feeding and to evoke neuroactivation in the PVN (via c_Fos staining) should be addressed.*

Indeed, we have already performed experiments looking to establish the potential involvement of NDN in the anorexigenic effects induced by leptin. These results were originally present in the manuscript as Figure 5. Additionally, we have performed new experiments comparing the effectiveness of LV-1075 and HS024 in blocking the anorexigenic effect of a central injection of leptin. Results presented in Figure 5.